# Risk factors relate to the variability of health outcomes as well as the mean: A GAMLSS tutorial

David Bann[1]*, Liam Wright[1], Tim J Cole[2]

[1]Centre for Longitudinal Studies, Social Research Institute, University College London, London, United Kingdom; [2]Great Ormond Street Institute of Child Health, University College London, London, United Kingdom

**Background:** Risk factors or interventions may affect the variability as well as the mean of health outcomes. Understanding this can aid aetiological understanding and public health translation, in that interventions which shift the outcome mean and reduce variability are typically preferable to those which affect only the mean. However, most commonly used statistical tools do not test for differences in variability. Tools that do have few epidemiological applications to date, and fewer applications still have attempted to explain their resulting findings. We thus provide a tutorial for investigating this using GAMLSS (Generalised Additive Models for Location, Scale and Shape).

**Methods:** The 1970 British birth cohort study was used, with body mass index (BMI; N = 6007) and mental wellbeing (Warwick-Edinburgh Mental Wellbeing Scale; N = 7104) measured in midlife (42–46 years) as outcomes. We used GAMLSS to investigate how multiple risk factors (sex, childhood social class, and midlife physical inactivity) related to differences in health outcome mean and variability.

**Results:** Risk factors were related to sizable differences in outcome variability—for example males had marginally higher mean BMI yet 28% lower variability; lower social class and physical inactivity were each associated with higher mean and higher variability (6.1% and 13.5% higher variability, respectively). For mental wellbeing, gender was not associated with the mean while males had lower variability (–3.9%); lower social class and physical inactivity were each associated with lower mean yet higher variability (7.2% and 10.9% higher variability, respectively).

**Conclusions:** The results highlight how GAMLSS can be used to investigate how risk factors or interventions may influence the variability in health outcomes. This underutilised approach to the analysis of continuously distributed outcomes may have broader utility in epidemiologic, medical, and psychological sciences. A tutorial and replication syntax is provided online to facilitate this (https://osf.io/5tvz6/).

**Funding:** DB is supported by the Economic and Social Research Council (grant number ES/M001660/1), The Academy of Medical Sciences / Wellcome Trust ("Springboard Health of the Public in 2040" award: HOP001/1025); DB and LW are supported by the Medical Research Council (MR/V002147/1). The funders had no role in study design, data collection and analysis, decision to publish, or preparation of the manuscript.

*For correspondence: david.bann@ucl.ac.uk

Competing interest: The authors declare that no competing interests exist.

## Editor's evaluation

Using data from the 1970 British Birth Cohort study, the authors demonstrated the utility of Generalized Additive Models for Location, Scale and Shape (GAMLSS) to investigate the association of three risk factors (sex, socioeconomic circumstances, and physical inactivity) with body mass index and mental wellbeing. This work provides empirical evidence for why we should consider how risk factors influence the variability and not just the mean of outcomes. From the perspective of developing

personalized medicine, it is important to know whether interventions have response heterogeneity as the first step. If such heterogeneity is identified, the next step will be to identify the factors associated with the heterogeneity (or those who will be benefitted from the intervention). Therefore, this study contributes to the first step by investigating the possibility of response heterogeneity.

## Introduction

What is health? Contrary to simplistic notions of its being defined as the absence of disease, it is now increasingly understood that most outcomes of public health significance are continuous in nature (*Keyes and Galea, 2016*). This applies to both physical and mental health outcomes (*Plomin et al., 2009*; *Keyes, 2002*). The use of binary endpoints, while having utility in clinical applications, should not hinder investigation of the influences of health outcomes which are ultimately continuous. Further, analysing the determinants of health using continuous rather than binary outcomes is beneficial both practically (with more statistical power and less information loss) and substantively (greater aetiological understanding). Indeed, those at high risk of a developing an illness may comprise a minority of those who ultimately succumb (*Rose, 2001*).

Studies into the effect on continuous outcomes of exposures, be they risk factors in observational studies or interventions in randomised trials, typically focus on mean differences in the outcome, using linear regression. However linear regression assumes homoscedasticity, that is that the variability of the outcome is unrelated to the exposure, and often this is not the case. It is possible to extend regression analysis to model the variability as well as the mean, and this has benefits in terms of not only the model's fit but also its interpretation. If for example the intervention in a trial can be shown to reduce variability in the outcome, this could reasonably be viewed as evidence of intervention success (*Subramanian et al., 2018*) independent of the intervention's effect on the mean. Treatment for refractive vision errors—glasses, contact lenses, and/or corrective surgery—seeks to improve vision by shifting individuals towards a specified standard (e.g. 20/20 vision) (*Vitale et al., 2006*). Successful treatments alter the mean refraction, but they are even more successful if they also reduce the substantial variability in refraction arising from the mix of short- and long-sighted individuals.

Similarly, obesity interventions aim to reduce body mass index (BMI) and shift treated individuals from overweight (25–30 kg/m$^2$), obese ( > 30 kg/m$^2$), or severely obese ( > 45 kg/m$^2$) to the normal range (20–25 kg/m$^2$). However, here the effect of the intervention on variability is often to increase it. Even if not formally tested, visual comparisons of outcome distributions of some influential trials suggest that weight loss interventions increase rather than reduce BMI variability, (*Truby et al., 2006*) presumably since they are effective in some but not all participants.

Understanding if and how risk factors influence variability in health outcomes has aetiological significance, consistent with the goal of epidemiological science to understand the *distribution* of health (*Porta, 2008*). Risk factors could feasibly affect outcome variability yet not affect the mean—for example, one study found that breastfeeding was not related to mean childhood BMI, yet was related to lower childhood BMI variability (*Beyerlein et al., 2008b*). Similarly, sex may affect variability and/or average levels of an outcome—for instance, males may have greater variability than females in some cognitive traits (*Hyde, 2014*) and brain structures (*Wierenga et al., 2022*).

Identifying associations between risk factors and outcome variability may also be useful to identify the absence or presence of heterogeneity in susceptibility to interventions or risk factors and thus aid aetiological understanding. Indeed, the finding that substantial increases in mean BMI in recent decades have been matched by increases in BMI variability indicates that there may be differential susceptibility to the obesogenic environment (*Flegal and Troiano, 2000*; *Johnson et al., 2015*). In the context of randomised controlled trials, the finding of variability in treatment effects between individuals has been used to justify individualised approaches to treatment (personalised medicine). Reflecting the challenges of empirically testing this, however, five separate meta-analyses have tested heterogeneity in response to antidepressant therapy; despite using the same dataset, different methods and divergent conclusions were drawn (*Luedtke and Kessler, 2021*).

Another advantage of modelling varability arises in common situations where the outcome under study is non-linearly related to other outcomes of interest. For instance, BMI influences mortality and morbidity rates, but the relationship between BMI and mortality is thought to be J-shaped ( *Bhaskaran et al., 2018*) compared with those in the normal range, mortality risks are greater for those who

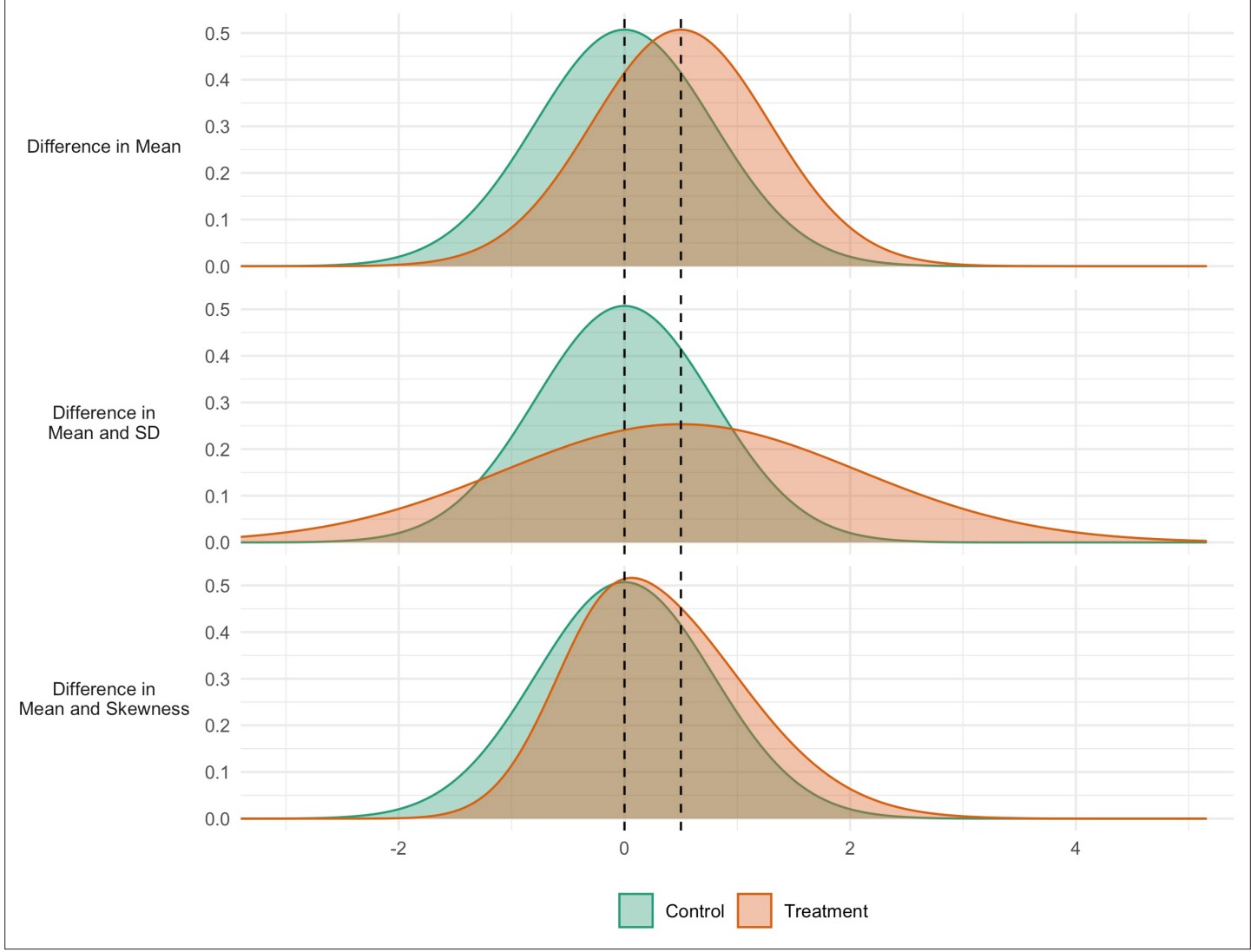

**Figure 1.** Simulated data for three interventions each having the same effect on the mean, but different effects on the variability (middle panel) and skewness (bottom panel).

are under- or overweight. In this case, the total effect of an intervention to reduce BMI on these wider outcomes is not fully captured by its average BMI effect. Rather, understanding the total distributional effect on BMI is required.

*Figure 1* shows three hypothetical scenarios for an intervention to affect the distribution of an outcome. In the first case (Panel A), the intervention has an impact that is consistent across the population: all individuals are affected and to the same extent. In the second case (Panel B), the intervention has the same mean impact, but variability is also increased: some are positively affected, others negatively. In the third case (Panel C), the mean is again increased, but so is skewness. There is heterogeneity in response, with some seeing more positive responses than others. The policy implications may be different in each case. In the second and third scenarios, efforts could be directed to identify those who are (more) positively impacted, so as to increase the net benefit or cost-effectiveness of the intervention. Indeed, in a choice between interventions, an intervention generating lower expected benefits but smaller variability in outcomes may be chosen, in so far as reducing inequalities is seen as a policy goal in itself.

Recent studies in biological (*Sun et al., 2020*; *Nakagawa et al., 2014*), environmental (*Pitt et al., 2020*), and economic science (*Hohberg et al., 2020*; *Silbersdorff and Schneider, 2019*; *Silbersdorff et al., 2018*) have begun to examine how risk factors relate to the distribution of the outcome

of interest. However, there have been few epidemiological applications of this approach to date; (*Beyerlein et al., 2008a*) and fewer still that provide explanations for such findings, which are essential if such methods are to have utility. Indeed, one recent study which investigated the association between mental health symptoms and lower income explicitly avoided interpretation of its findings on variability, focusing instead on issues relating to the application of such methods (*Silbersdorff and Schneider, 2019*).

Regression methods that allow variability to be modelled are uncommon. One particular method, Generalised Additive Models for Location, Scale and Shape (GAMLSS) (*Rigby and Stasinopoulos, 2005*) has become the standard for constructing growth reference centiles, (*Cole et al., 2009*) where the aim is to model the outcome's distribution as a function of age. It defines the distribution in terms of distribution moments, i.e. the mean, variance, and optionally skewness and kurtosis. This allows for factors influencing the higher moments to be identified in just the same way as for the mean, and it provides a simple and elegant interface for modelling variability in epidemiology.

Another arguably underutilised (*Beyerlein, 2014*) and related statistical approach to investigating risk factors for continuous outcomes is quantile regression. Recent epidemiological studies using this method have found that risk factors for higher BMI—particularly lower social class and physical inactivity—have sizably larger effect sizes at higher BMI centiles (*Bann et al., 2020*; *Green and Rowe, 2020*). This has potentially important policy implications—risk factors which have larger effects amongst those at highest health risk are likely to have a more favourable effect on population health than alternatives which do not (*Bann et al., 2020*). However, the reason for this phenomenon is not yet understood—it is likely to be logically consistent with results of GAMLSS analyses in which risk factors influence outcome means, variability and/or skewness.

In this paper, we provide a worked example of the use and interpretation of GAMLSS. Accompanying this is an online tutorial and full replication syntax for running GAMLSS in R (https://osf.io/5tvz6/). We investigate whether and how several established risk factors—sex, childhood socioeconomic circumstances, and physical inactivity (*Stringhini et al., 2017*)—relate to differences in outcome mean and variability. We choose two different continuous outcomes, an indicator of adiposity (body mass index, BMI) and mental wellbeing. These are two weakly correlated health outcomes, each of independent importance to population health. Each risk factor-outcome combination is the subject of previous (separate) literature which focuses largely on mean differences only. For instance, low socioeconomic position in childhood has been repeatedly related to higher BMI (*Bann et al., 2018*; *Senese et al., 2009*) and worse mental wellbeing in adulthood; (*Wood et al., 2021*; *Simanek et al., 2021*; *Wood et al., 2017*) greater physical activity has notable likely bi-directional links with lower BMI ; (*Jakicic et al., 2019*) and higher wellbeing; (*Black et al., 2015*; *Choi et al., 2019*; *Pinto Pereira et al., 2014*) while males and females seemingly have similar mean BMI and wellbeing, (*Wood et al., 2017*) this may mask differences in variability or skewness, as suggested in the sizable sex differences in overweight and obesity rates (*Conolly et al., 2017*).

The further investigation of differences in variability and skewness in these outcomes is therefore arguably of substantive interest, providing further motivation to the tutorial content. We highlight the contribution of GAMLSS by contrasting results with the more commonly used linear regression and (less commonly used) quantile regression models.

## Methods

### Study sample

The 1970 British birth cohort study consists of all 17,196 babies born in Britain during one week of March 1970, with 9 subsequent waves of follow-up from childhood to midlife (*Elliott and Shepherd, 2006*) At the most recent wave (46 years), 12,368 eligible participants (those alive and not lost to follow-up) were invited to be interviewed at home by trained research staff—8581 participants provided at least some data in this wave. At all waves, informed consent was provided and ethical approval granted.

### Health outcomes

We selected two outcomes in midlife which capture different dimensions of health and are continuously distributed: adiposity (BMI), and mental wellbeing (Warwick-Edinburgh Mental Wellbeing Scale

(WEMWBS)). BMI was measured at 46 years, and wellbeing at 42 years (*Wood et al., 2021*) WEMWBS consists of 14 positively worded items—such as "I've been feeling optimistic about the future" and "…feeling cheerful"—measured on a five-point Likert scale, which are summed to give a total well-being score ranging from 14 to 70 (highest well-being) (*Tennant et al., 2007*).

## Risk factors

We chose three risk factors across different domains—each of them likely to independently influence health outcomes (*Stringhini et al., 2017*). They were coded as binary variables to simplify comparison of descriptive and GAMLSS results: sex (female/male), socioeconomic position (social class at birth; coded as non-manual/manual), and a behavioural risk factor (reported physical activity at 42 years; reported days in which the participant took part in exercise for 30 min or more in a typical week 'working hard enough to raise your heart rate and break into a sweat', coded as active ( ≥ 1 days)/ inactive (0 days)). We examined if the binary split of risk factors influenced the inferences drawn— additional analyses were conducted with them coded instead as categorical variables (social class in six categories and physical inactivity from 0 to 7 days).

## Analytical strategy

To visually inspect the outcome distributions and their differences across risk factor groups, we first plotted separate kernel density estimates alongside relevant descriptive statistics (mean, standard deviation, and coefficient of variation [CoV = SD/mean]). This enables a descriptive depiction of variability, with unadjusted GAMLSS results corresponding to each descriptive statistic. We then used GAMLSS (*Rigby and Stasinopoulos, 2005*) separately with each outcome, to formally investigate whether risk factors were associated with (1) differences in mean outcome, (2) differences in outcome variability, and (3) differences in outcome skewness. Linear regression analysis, in contrast, only enables mean differences in outcomes to be investigated.

GAMLSS is a form of regression analysis that estimates different 'moments' of the outcome distribution. The first moment is the location (see mean in *Figure 1* panel a), the second is variance, which specifies the scale or spread (SD in *Figure 1* panel b) the third is skewness which quantifies the relative size of the distribution tails (*Figure 1* panel c). As in linear regression analyses covariates can optionally be included, and appropriate link functions can be chosen for use.

GAMLSS requires that the distribution is specified at the outset. In this tutorial we use two distributions which we recommend for use in epidemiological research of continuous outcomes. First, the normal distribution (called NO in GAMLSS), where location is measured by the mean and scale by the standard deviation (SD). The normal distribution has no 'shape' moments, as there is no skewness and kurtosis is fixed.

Second, a more complex distribution which enables skewness to be investigated: the Box-Cox Cole and Green (BCCG). Here location is the median, scale is the generalised coefficient of variation (CoV), which is calculated in the normal case as SD/mean, and shape is skewness as defined by the Box-Cox power required to transform the outcome distribution to normality. The transformation requires the outcome to be on the positive line, so zero or negative values are excluded. BCCG is effectively NO with added skewness, though parameterised differently. A Box-Cox power of 1 indicates that the distribution is normal, 0 is log-normal and –1 inverse normal, so a smaller (i.e. more negative) power corresponds to more right skewness.

After choosing a distribution, linear models are used to specify the relationship between the independent variables and the different moments of the outcome distribution. As with other regression models, GAMLSS provides a standard error for each estimated coefficient, from which 95% confidence intervals can be calculated. We note that more experienced users may wish to use alternative distributions which GAMLSS facilitates (*Rigby et al., 2019*).

In our primary analyses we used the NO and BCCG families. Differences in variability are modelled with a log link, and can be multiplied by 100 and interpreted as percentage differences in variability to aid interpretation (*Lewontin, 1966*). Differences in the mean and median were also analysed as percentages, to aid comparability across outcomes and model estimates. To aid comparison of descriptive statistics and model estimation results, we first conducted analyses adjusting for each risk factor alone. We then adjusted for the risk factors jointly.

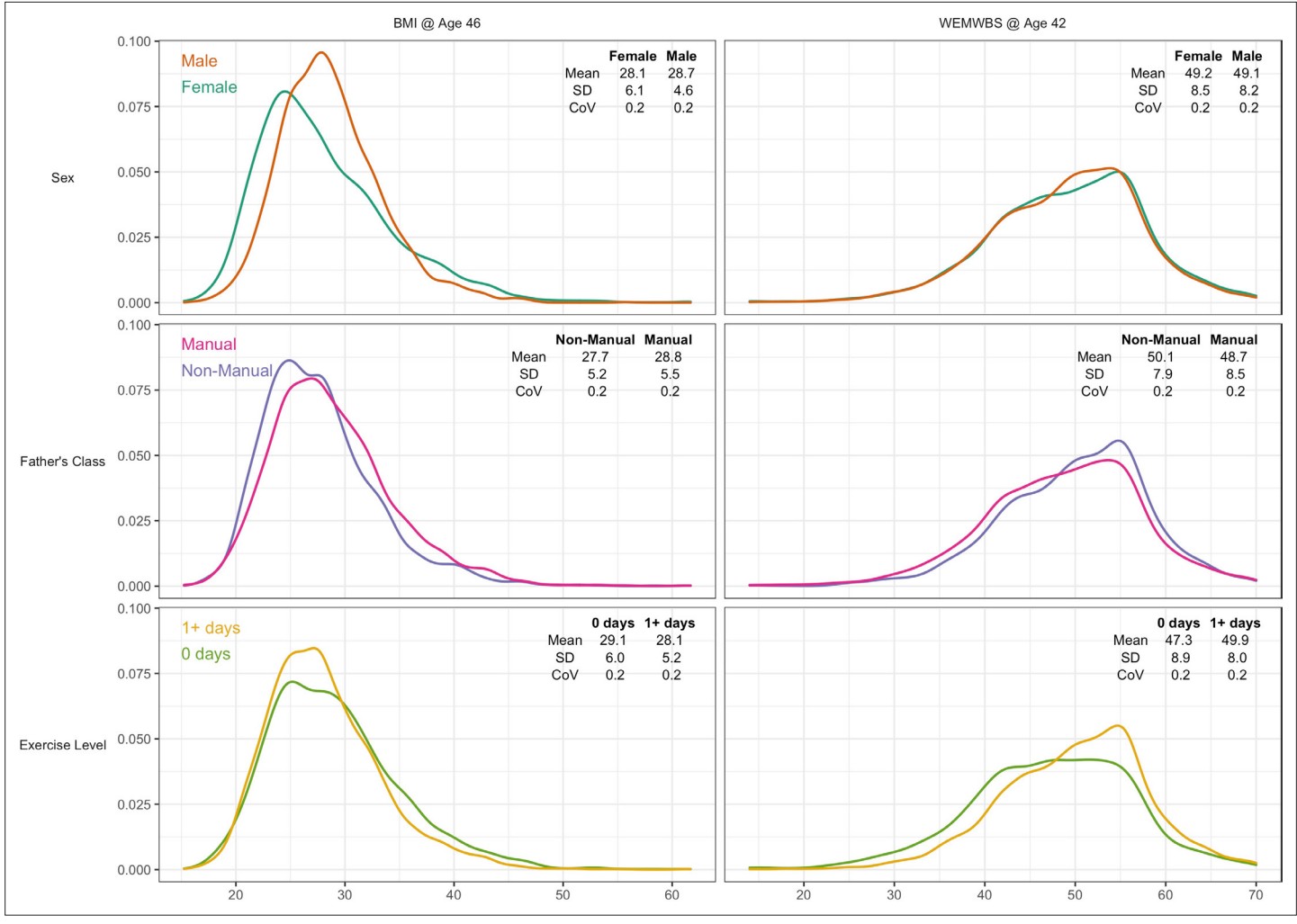

**Figure 2.** Kernel density plots for body mass index and mental wellbeing, stratified by risk factor group. Note: CoV = coefficient of variation (SD/mean).

Separately we fitted conditional quantile regression models to estimate risk factor and BMI associations at the lower, middle and upper quartiles of the outcome distribution, that is the 25th, 50th, and 75th centiles.

All analyses were conducted using R v4.1.1. We used the *gamlss* package version 5.3–4 to produce gamlss models (*Stasinopoulos and Rigby, 2007*). Syntax to replicate all analyses is presented online (https://osf.io/5tvz6/).

## Results

A total of 6007 participants had valid data for BMI and all risk factors, and 7104 for WEMWBS. Mean BMI was 28.4 (SD = 5.5), and mean WEMWBS 49.2 (8.3). Higher BMI was weakly associated with lower wellbeing ($r$ = –0.07, p < 0.01). BMI was moderately right-skewed (*Figure 2*, left panel) and WEMWBS left-skewed (*Figure 2*, right panel). Visual and descriptive comparisons of the BMI and wellbeing distributions by risk factor suggest that differences in the outcome mean and variability are not always in the same direction.

GAMLSS results for the binary risk factors are shown in *Tables 1 and 2*, with the results using the extra risk factor categories in*Supplementary file 1*. Associations were similar in the unadjusted and mutually adjusted analyses, so the former are described below.

**Table 1.** Risk factors in relation to body mass index: differences in mean, variability and skewness estimated by GAMLSS (n = 6007).

| Risk factor | % | NO distribution | | BCCG distribution | | |
|---|---|---|---|---|---|---|
| | | **Mean** | **SD** | **Median** | **CoV** | **Skewness*** |
| Female (ref) | 52.4% | 28.1 | 6.1 | 26.9 | 0.22 | 1.10 |
| Male | 47.6% | 28.7 | 4.6 | 28.2 | 0.16 | 0.75 |
| Unadjusted difference, % (SE) | | 1.9 (0.5) | –27.6 (1.8) | 4.1 (0.4) | –23 (1.8) | 0.48 (0.11) |
| Adjusted† difference, % (SE) | | 2.2 (0.5) | –27.4 (1.8) | 4.4 (0.4) | –22.6 (1.8) | 0.54 (0.11) |
| | | | | | | |
| Non-manual (ref) | 36.3% | 27.7 | 5.2 | 27 | 0.19 | 1.15 |
| Manual social class | 63.7% | 28.8 | 5.5 | 28 | 0.19 | 0.90 |
| Unadjusted difference, % (SE) | | 4.0 (0.5) | 6.1 (1.9) | 4.4 (0.5) | 6 (1.9) | 0.39 (0.11) |
| Adjusted† difference, % (SE) | | 3.8 (0.5) | 5.5 (1.9) | 4.3 (0.4) | 5.6 (1.9) | 0.40 (0.12) |
| | | | | | | |
| Physically active (ref) | 73% | 28.1 | 5.2 | 27.4 | 0.19 | 0.97 |
| Inactive | 27% | 29.1 | 6.0 | 28.3 | 0.21 | 0.94 |
| Unadjusted difference, % (SE) | | 3.3 (0.6) | 13.5 (2.1) | 2.9 (0.5) | 10.4 (2.1) | 0.08 (0.12) |
| Adjusted† difference, % (SE) | | 3.3 (0.6) | 12.1 (2.1) | 3.1 (0.5) | 9.3 (2.1) | 0.12 (0.12) |

*Skewness is estimated as the Box-Cox power (that is, the power required to transform the outcome to a normal distribution); differences are the absolute difference in Box-Cox power in each subgroup estimated by GAMLSS. GAMLSS estimates multiple distribution moments simultaneously; thus, differences may not exactly correspond to descriptive comparisons reported above.

†Estimates mutually adjusted for sex, social class and physical inactivity.

NO: normal distribution. BCCG: Box-Cox Cole and Green distribution: SD: standard deviation. CoV: coefficient of variation. GAMLSS: Generalized Additive Models for Location, Scale and Shape. SE, standard error.

## Body mass index

Males had higher mean BMI yet lower variability than females—see *Figure 2* and *Table 1*. The SD for BMI was lower in males (4.6) than females (6.1) that is a 28% difference (difference in log(SD) *100). This matches the estimate obtained from GAMLSS—males had 27.6% (SE: 1.8%) less variability than females (*Table 1*).

In contrast, lower social class and physical inactivity were both associated with higher mean BMI and higher BMI variability (*Figure 2* and *Table 1*). Those from lower social class households had 4% (SE 0.5%) higher mean BMI than those from non-manual classes, and 6.1% (1.9%) more variability. Physically inactive participants had 3.3% (0.6%) higher mean BMI and 13.5% (2.1%) more variability.

The GAMLSS results were similar with the BCCG distribution rather than NO (*Table 1*). That is, risk factors associated with higher mean BMI and higher SD were also associated with higher median BMI and higher CoV. Male sex and lower social class were both associated with less right skewness of the BMI distribution; the Box-Cox power was 0.5 (0.1) higher in males and 0.4 (0.1) higher for manual social class. Physical activity was not associated with outcome skewness.

## Mental wellbeing – Warwick-Edinburgh mental wellbeing scale

There was little evidence of sex differences in mean wellbeing, while males had marginally less variability than females by 3.9% (1.7%). Lower social class and physical inactivity were both associated with lower mean yet higher variability (*Figure 2* and *Table 2*). Those from lower social class households had a 2.8% (0.4%) lower mean yet 7.2% (1.8%) higher variability. Physically inactive participants had 5.3% (0.5%) lower mean yet 10.9% (1.9%) higher variability. These findings were similar in mutually adjusted analyses (*Table 2*).

**Table 2.** Risk factors in relation to mental wellbeing (WEMWBS): differences in mean, variability and skewness estimated by GAMLSS (n = 7,104).

| Risk factor | % | NO distribution | | BCCG distribution | | |
| --- | --- | --- | --- | --- | --- | --- |
| | | Mean | SD | Median | COV | Skewness* |
| Female (ref) | 52.8% | 49.2 | 8.5 | 50 | 0.17 | –0.41 |
| Male | 47.2% | 49.1 | 8.2 | 50 | 0.17 | –0.40 |
| Unadjusted difference, % (SE) | | –0.2 (0.4) | –3.9 (1.7) | –0.3 (0.4) | –3.5 (1.7) | 0.02 (0.11) |
| Adjusted† difference, % (SE) | | –0.6 (0.4) | –3.6 (1.7) | –0.7 (0.4) | –2.6 (1.7) | 0.00 (0.11) |
| | | | | | | |
| Non-manual (ref) | 34.8% | 50.1 | 7.9 | 51 | 0.16 | –0.45 |
| Manual social class | 65.2% | 48.7 | 8.5 | 49 | 0.17 | –0.37 |
| Unadjusted difference, % (SE) | | –2.8 (0.4) | 7.2 (1.8) | –2.9 (0.4) | 10.9 (1.8) | –0.20 (0.12) |
| Adjusted† difference, % (SE) | | –2.5 (0.4) | 6.0 (1.8) | –2.7 (0.4) | 9.8 (1.8) | –0.24 (0.12) |
| | | | | | | |
| Physically active (ref) | 72.4% | 49.9 | 8.0 | 51 | 0.16 | –0.38 |
| Inactive | 27.6% | 47.3 | 8.9 | 48 | 0.19 | –0.36 |
| Unadjusted difference, % (SE) | | –5.3 (0.5) | 10.9 (1.9) | –5.2 (0.4) | 16.2 (1.9) | –0.12 (0.12) |
| Adjusted† difference, % (SE) | | –5.3 (0.5) | 9.9 (1.9) | –5.1 (0.4) | 15.2 (1.9) | –0.10 (0.12) |

*Skewness is estimated as the Box-Cox power (that is, the power required to transform the outcome to a normal distribution); differences are the absolute difference in Box-Cox power in each subgroup estimated by GAMLSS. GAMLSS estimates multiple distribution moments simultaneously; thus, differences may not exactly correspond to descriptive comparisons reported above.

†Estimates mutually adjusted for sex, social class and physical inactivity.

NO: normal distribution. BCCG: Box-Cox Cole and Green distribution: SD: standard deviation. CoV: coefficient of variation. GAMLSS: Generalized Additive Models for Location, Scale and Shape. SE, standard error.

 The results were similar with the BCCG distribution (*Table 2*). There was evidence suggesting that lower social class was associated with less skewness in the wellbeing distribution; sex and physical activity were not associated with outcome skewness.

## Comparison with quantile regression findings

For BMI, the associations of lower social class and physical inactivity were stronger at upper quantiles (*Table 3*; e.g., manual social class had 3.7 (0.6) higher BMI at the the median, and 4.9 (0.7) at the 75th); estimates at higher centiles were also estimated less precisely than at lower centiles (larger SE). In

**Table 3.** Risk factors in relation to body mass index (BMI) and mental wellbeing (WEMWBS): percentage differences at multiple points of the outcome distribution estimated by quantile regression.

| Outcome | Risk factor | 25th centile | 50th centile | 75th centile |
| --- | --- | --- | --- | --- |
| | Male vs female | 6.8 (0.5) | 4.5 (0.6) | –0.8 (0.7) |
| | Father's Class | 3.7 (0.6) | 3.7 (0.6) | 4.9 (0.7) |
| BMI @ Age 46 | Exercise Level | 1 (0.7) | 3 (0.7) | 4.3 (0.8) |
| | Sex | 0 (0.7) | 0 (0.5) | 0 (0.3) |
| | Father's Class | –4.5 (0.7) | –4 (0.5) | –1.8 (0.3) |
| WEMWBS @ Age 42 | Exercise Level | –6.9 (0.5) | –6.1 (0.5) | –1.8 (0.5) |

Note: results show the percentage difference (log-transformed x 100) in BMI or mental wellbeing (WEMWEBS; standard errors in parenthesis) at different centiles of the outcome distribution; estimates are mutually adjusted.

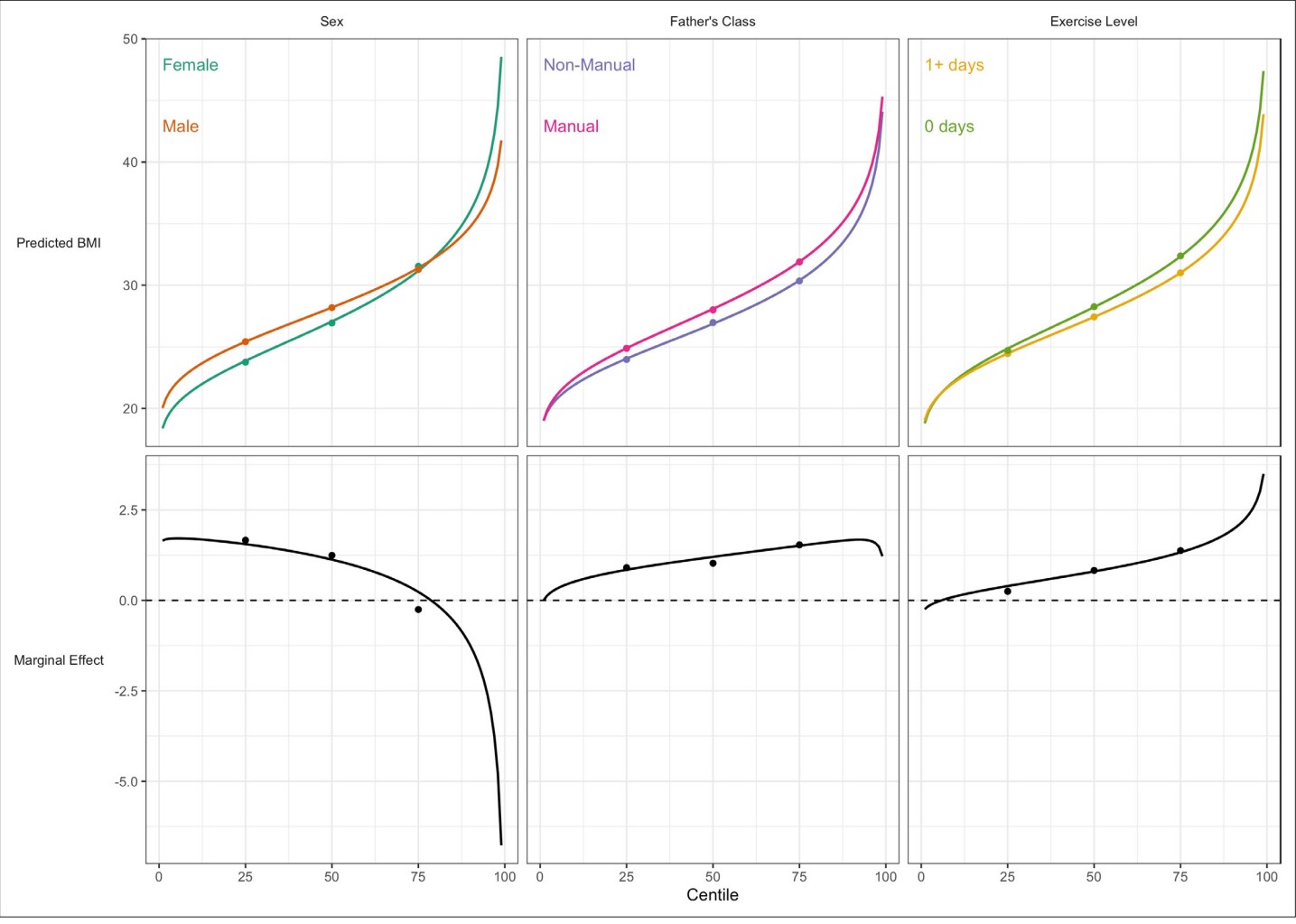

**Figure 3.** Association between risk factors and BMI by BMI centile. Plotted lines are calculated using GAMLSS estimation results of the entire outcome distribution; points at the 25th, 50th, and 75th centiles are estimated using quantile regression models. Marginal effects show the differences in outcome between each risk group across the outcome distribution.

contrast sex differences were present at lower centiles but absent at the 75th centile. These findings corresponded with those from GAMLSS using BCCG, with all BMI centiles plotted by risk factor group (*Figure 3*). This comparison highlights the utility of GAMLSS—risk factor differences in the mean, variability, and skewness can each be quantified and thus visually depicted.

For WEMWBS, the associations of lower social class and physical inactivity were also stronger at lower quantiles (*Table 3*), yet had larger standard errors. Sex was not associated with WEMWBS at any centile. These findings corresponded with those from GAMLSS (*Figure 4*).

## Discussion

Using an underutilised analytical approach (GAMLSS), we present empirical evidence to support the idea that risk factors can relate to sizable differences in outcome variability, and even outcome skewness, in addition to differences in the outcome mean. Females had higher variability in BMI and mental wellbeing than males; lower social class and physical inactivity were each associated with higher variability in both BMI and mental wellbeing, despite having different directions of association with the mean (higher BMI yet lower mental wellbeing).

Our findings add to an emerging literature which has investigated associations between risk factors and outcome variability. Studies (*Sun et al., 2020*; *Nakagawa et al., 2014*; *Pitt et al., 2020*; *Hohberg et al., 2020*; *Silbersdorff and Schneider, 2019*; *Silbersdorff et al., 2018*; *Beyerlein et al., 2008a*)

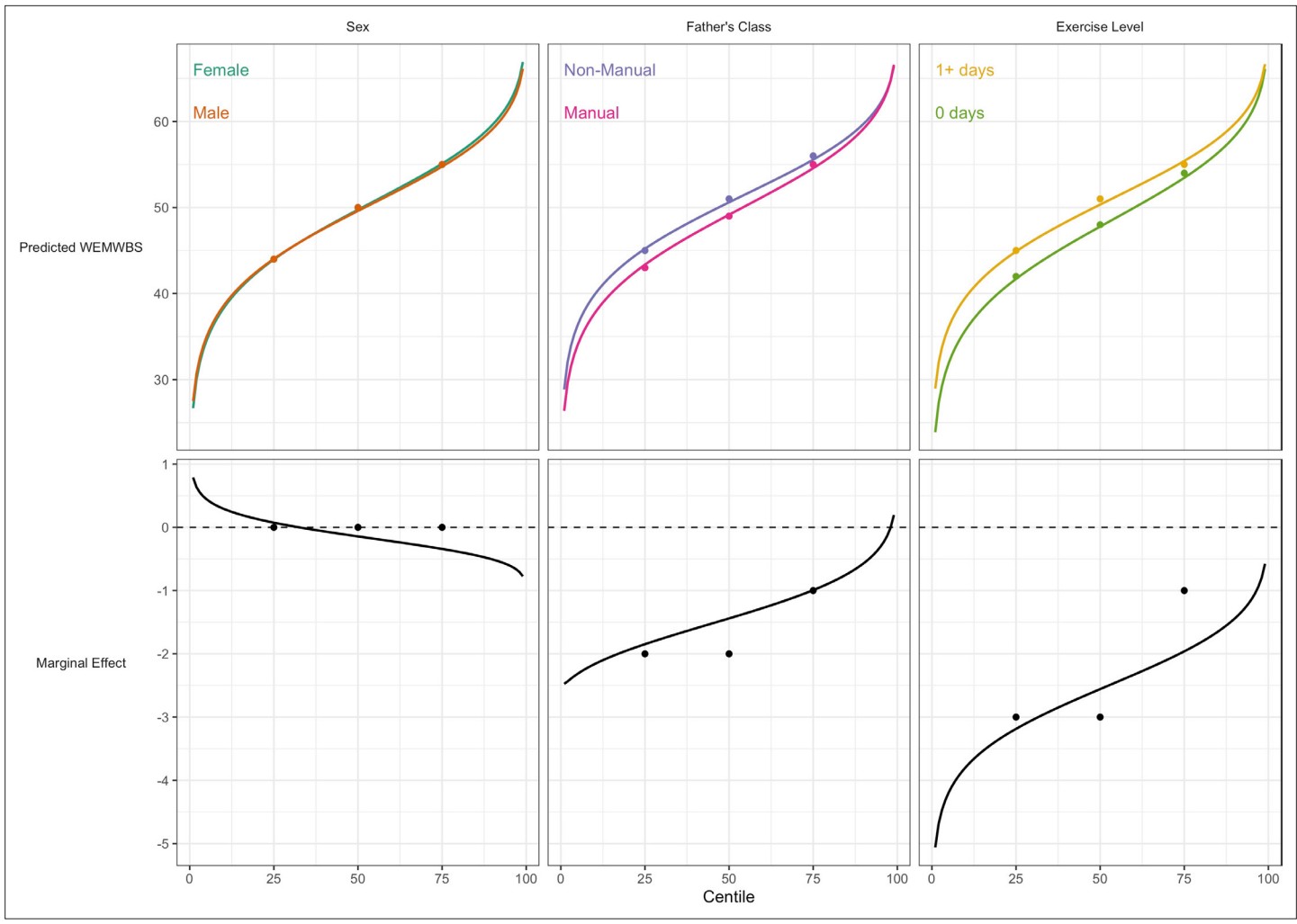

**Figure 4.** Association between risk factors and mental wellbeing (WEMWBS) by centile. Plotted lines are calculated using GAMLSS estimation results of the entire outcome distribution; points at the 25th, 50th, and 75th centiles are estimated using quantile regression models. Marginal effects show the differences in outcome between each risk group across the outcome distribution.

have reported that risk factors associated with higher means are also associated with higher outcome variability. For example, (*Beyerlein et al., 2008a*) found that multiple risk factors for high childhood BMI (such as more frequent television viewing and greater rapid infant weight gain) were related to both higher mean BMI and greater variability in BMI. However, previous studies have not utilised multiple outcomes or nationally representative samples, and have not systematically considered explanations for such findings or their implications.

Our findings help to reconcile findings from GAMLSS with those using quantile regression (*Beyerlein et al., 2008a*; *Bann et al., 2020*; *Green and Rowe, 2020*) which have reported stronger effect sizes for BMI risk factors at higher BMI centiles. This finding is both consistent with and helps explain the GAMLSS findings. For instance, lower social class and physical inactivity are related to higher BMI mean and variability, yet less BMI skewness; the net result is higher effect estimates at upper centiles which are less precisely estimated, as seen in quantile regression. While both analytical approaches have merit, GAMLSS has a number of attractive features for use in aetiological research: it enables each distribution moment to be separately investigated, and uses predetermined distribution families which enable computation of sparsely distributed variables.

Why are risk factors associated with differences in outcome variability? There are multiple possible explanations. First, risk factors may not be sufficient for an outcome to occur but rather only have a causal effect in the presence of other factors, for instance as posited in models such as the *stress-diathesis* model of mental health (*Zuckerman, 1999*). Such additional factors could also operate as

effect modifiers which increase the strength of the risk factor. Factors such as genetic propensity to weight gain may for example modify the effect on weight gain of exposure to adverse socioeconomic circumstances (*Tyrrell et al., 2017*). Other environmental factors could operate similarly—such that the association between lower social class and higher BMI is weaker amongst those living in a local environment which is less 'obesogenic' (i.e. less conducive to physical inactivity and lower energy intake) (*Drewnowski et al., 2007*; *Stafford et al., 2007*). The net result of such divergent effects would be increased variability since the effects would range from zero to the upper bound of the effect. This explanation may also apply to mental wellbeing, given evidence for the myriad environmental (*Ludwig et al., 2012*; *Wood et al., 2021*) and genetic determinants (*Luciano et al., 2018*; *de Moor et al., 2015*) which could modify the effects observed in the current study.

Alternatively, between-person differences in confounding and/or measurement error may also lead to risk factors being associated with outcome variability. For example, in the present study physical activity was measured via a single item capturing reported activity of a moderate-vigorous intensity for at least 30 min per day; this is an imperfect reflection of the underlying exposure which may have a causal effect (e.g. total energy expenditure [across all intensities of activity] in the case of adiposity; (*Bann et al., 2014*) or time spent in specific activities conducive to wellbeing in the case of mental wellbeing [*Black et al., 2015*]). The net result would be higher variability in those reporting higher physical activity levels. A related issue is the extent to which the exposure captures the same 'dose' across participants in a given study. The physical activity measure used here counted the number of days that bouts of activity lasted at least 30 min; this likely reflects substantial variability in the level of exercise actually undertaken, thus leading to greater differences in outcome variability. This could partly explain the associations of lower social class with greater outcome variability, since social class is one dimension of socioeconomic position, such that there may be substantial between-person variation in other dimensions (e.g. parental education, income and/or wealth [*Moulton et al., 2021*; *Galobardes et al., 2006*]) which may each influence outcomes, leading to greater variability.

The study highlights the fact that analyses by GAMLSS and quantile regression lead to similar results at the selected quantiles of the outcome distribution—see *Figures 3 and 4*. However GAMLSS, by analysing the whole distribution, can in some cases provide more efficient estimates of the quantiles. Compare for example the standard errors of the median as obtained by the BCCG distribution (*Tables 2 and 3*) and quantile regression (*Table 3*); the GAMLSS standard errors are smaller.

## Strengths and limitations

Strengths of this study include the analytical approach used (GAMLSS) to empirically investigate differences in outcome variability. While differences in variability can be informed by descriptive comparison (e.g. comparing standard deviations), GAMLSS additionally enables computation of estimates of precision and incorporates multivariable specifications (e.g. confounder or mediator adjustment; and inclusion of interaction terms). The use of the 1970 birth cohort data is an additional strength, enabling investigation of multiple risk factors and two largely orthogonal yet important continuous health outcomes. The national representation of this cohort is also advantageous—highly distorted sample selection can bias conventional epidemiological results (i.e. mean differences in outcomes) (*Munafò et al., 2018*), and may also bias comparisons of outcome variability.

The study also has limitations. As in all observational studies, causal inference is challenging despite the use of longitudinal data. Associations of social class at birth with outcomes for example could be explained by unmeasured confounding—this may include factors such as parental mental health. This is challenging to falsify empirically owing to a lack of such data collected before birth. In contrast, sex is randomly assigned at birth, and thus its associations with outcomes are unlikely to be confounded. However, sex differences in reporting may bias associations with mental wellbeing. Physical activity and mental wellbeing were ascertained at broadly the same age, so that associations between the two could be explained by reverse causality; existing evidence appears to suggest bi-directionality of links between physical activity and both outcomes (*Pinto Pereira et al., 2014*; *Barone Gibbs et al., 2020*). Finally, attrition led to lower power to precisely estimate smaller effect sizes (e.g. gender differences in mental wellbeing) or confirm null effects. Such attribution could potentially bias associations—those in worse health and adverse socioeconomic circumstances are disproportionately lost to follow-up (*Mostafa and Wiggins, 2015*; *Mostafa et al., 2021*). The focus of principled approaches to handle missing data in epidemiology has been on the main parameter of interest—typically beta coefficients

in linear regression models—and further empirical work is required to investigate the potential implications of (non-random) missingness for the variability and other moments of the outcome distribution.

## Potential implications

This study used an underutilised approach to empirically investigate associations between risk factors and outcome variability in a single cohort study. Thus, our findings require replication and extension in other datasets across other risk factors and health outcomes. Future studies should also seek to explain their findings, and where possible falsify potential explanations. Understanding how risk factors relate to and/or cause differences in outcome variability is not a standard part of epidemiological training, and it entails additional analytical and conceptual complexity. Thus, with greater application of these tools an emerging consensus on best practice should develop. In the first instance, we recommend both descriptive and formal investigation, and that analysts carefully consider the use of both absolute (e.g. SD) and relative (e.g. CoV) differences in variability. Since the CoV is fractional standard deviation (e.g. SD/mean or log SD), its suitability of use depends on the a priori anticipated relationship between the mean and variance.

In the context of randomised controlled trials, the finding of variability in treatment effects between individuals has been used to justify individualised approaches to treatment (personalised medicine). It is beyond the scope of the current article to discuss the tractability of this for complex outcomes in which treatment effects are unpredictable (*Davey Smith, 2011*). Trials are designed typically to detect only mean differences in outcomes (*Senn, 2016*); nevertheless, additionally presenting outcome variability before and after treatment would be helpful to better appraise intervention effects (*Subramanian et al., 2018*). GAMLSS provides a useful framework with which to formally investigate this, even where the homoscedasticity assumption does not hold (i.e. where risk factors or treatment groups differ in their outcome variance). Where there are multiple potential efficacious interventions, further studies could meta-analyse existing trials to identify the types of intervention which additionally reduce outcome variability.

## Conclusion

We provide empirical support for the notion that risk factors or interventions can either reduce or increase variability in health outcomes. This finding is consistent with results from quantile regression analysis where a risk factor vs outcome association is stronger (or weaker) at higher outcome centiles. Such findings may be explained by heterogeneity in the causal effect of each exposure, by the influence of other (typically unmeasured) variables, and/or by measurement error. This underutilised approach to the analysis of continuously distributed outcomes may have broader utility in epidemiological, medical, and psychological sciences. Our tutorial and syntax content is designed to facilitate this.

## Additional information

### Funding

| Funder | Grant reference number | Author |
| --- | --- | --- |
| Medical Research Council | MR/V002147/1 | David Bann<br>Liam Wright |
| Economic and Social Research Council | ES/M001660/1 | David Bann |
| Wellcome Trust | HOP001/1025 | David Bann |

The funders had no role in study design, data collection and interpretation, or the decision to submit the work for publication.

### Author contributions

David Bann, Conceptualization, Data curation, Formal analysis, Funding acquisition, Investigation, Methodology, Resources, Visualization, Writing - original draft, Writing - review and editing; Liam Wright, Formal analysis, Investigation, Methodology, Resources, Software, Visualization, Writing

- review and editing; Tim J Cole, Conceptualization, Investigation, Methodology, Visualization, Writing - review and editing

### Author ORCIDs
David Bann (iD) http://orcid.org/0000-0002-6454-626X

### Ethics
Human subjects: This paper uses secondary data analysis using data from a cohort study which has been followed-up since birth in 1970. Cohort members provided informed consent, and the study received full ethical approval - most recently from the NRES Committee South East Coast-Brighton and Sussex.

### Decision letter and Author response
Decision letter https://doi.org/10.7554/eLife.72357.sa1
Author response https://doi.org/10.7554/eLife.72357.sa2

---

## Additional files

### Supplementary files
• Supplementary file 1. Risk factors in relation to body mass index (BMI): differences in mean, variability and skewness estimated by GAMLSS (b) Risk factors in relation to mental wellbeing (WEMWEBS): differences in mean, variability and skewness estimated by GAMLSS.

• Transparent reporting form

### Data availability
All data are available to download from the UK Data Archive: https://beta.ukdataservice.ac.uk/datacatalogue/series/series?id=200001.

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
