## [Editor Report]

Using data from the 1970 British Birth Cohort study, the authors demonstrated the utility of Generalized Additive Models for Location, Scale and Shape (GAMLSS) to investigate the association of three risk factors (sex, socioeconomic circumstances, and physical inactivity) with body mass index and mental wellbeing. This work provides empirical evidence for why we should consider how risk factors influence the variability and not just the mean of outcomes. From the perspective of developing personalized medicine, it is important to know whether interventions have response heterogeneity as the first step. If such heterogeneity is identified, the next step will be to identify the factors associated with the heterogeneity (or those who will be benefitted from the intervention). Therefore, this study contributes to the first step by investigating the possibility of response heterogeneity.

---

## [Decision Letter]

**Decision letter after peer review:**

Thank you for submitting your article "Risk factors relate to the variability of health outcomes as well as the mean" for consideration by *eLife*. Your article has been reviewed by 3 peer reviewers, and the evaluation has been overseen by a Reviewing Editor and a Senior Editor. The following individual involved in review of your submission has agreed to reveal their identity: Carmen Tekwe (Reviewer #3).

As is customary in *eLife*, the reviewers have discussed their critiques with one another. What follows below is the Reviewing Editor's edited compilation of the essential and ancillary points provided by reviewers in their critiques and in their interaction post-review. Please submit a revised version that addresses these concerns directly. Although we expect that you will address these comments in your response letter, we also need to see the corresponding revision clearly marked in the text of the manuscript. Some of the reviewers' comments may seem to be simple queries or challenges that do not prompt revisions to the text. Please keep in mind, however, that readers may have the same perspective as the reviewers. Therefore, it is essential that you attempt to amend or expand the text to clarify the narrative accordingly.

Essential revisions:

The authors claim that the primary aim of this work is "exploring factors affecting outcome variability in an epidemiological context." This aim seems to be very broad, and it is unclear how one would address this aim in a single manuscript. We suggest defining the aims of the manuscript clearly in terms of the objectives the authors want to achieve. For instance, what would the audience gain by reading the manuscript (objective of a tutorial type of manuscript)? Or what is the research question the authors aim to investigate (objective of a non-tutorial type of manuscript)?

In the field of epidemiology, it is well understood that an exposure may change different parameters of the outcome distribution in the population (1). For example, a population intervention focusing only on a high-risk group would increase the right skewness of the outcome distribution in that population after implementation. Further, it is unclear how using a model that already assumes that independent variables may affect the variability of the outcome (by parameterizing this relationship) can alone provide empirical support for the that notion. Instead, having used such a model, the authors could report on the effect estimates of the risk factor on the variability of the outcome measures. In other words, more clarity is needed regarding the takeaway message of the manuscript.

We suggest that the authors make this manuscript a tutorial; if they agree with our suggestion, the following additions would considerably improve the manuscript:

i) Clearly annotated R and Stata codes to replicate the analysis. This would provide potential users of the proposed technique t with hands-on exercise.

ii) Clear examples of interpretation within epidemiological context. For example, how should one interpret the percentage point difference in SD and the uncertainty around it?

iii) Comparison between the results of GAMLSS and a technique that does not model the variance and further elaboration on the advantages of fitting this complex model over a simple model.

iv) Explanations answering the following questions: What do we learn from comparing the descriptive kernel density estimates to the unadjusted estimates? Are they supposed to be very similar? If yes, why?

v) Discussions on or recommendation for addressing the on challenges in choosing the type of outcome distribution in GAMLSS within epidemiological context.

(1) Rose G. Sick individuals and sick populations. Int J Epidemiol. 2001 Jun 1;30(3):427-32.

---

## [Author Response]

Essential revisions:The authors claim that the primary aim of this work is "exploring factors affecting outcome variability in an epidemiological context." This aim seems to be very broad, and it is unclear how one would address this aim in a single manuscript. We suggest defining the aims of the manuscript clearly in terms of the objectives the authors want to achieve. For instance, what would the audience gain by reading the manuscript (objective of a tutorial type of manuscript)? Or what is the research question the authors aim to investigate (objective of a non-tutorial type of manuscript)?In the field of epidemiology, it is well understood that an exposure may change different parameters of the outcome distribution in the population (1). For example, a population intervention focusing only on a high-risk group would increase the right skewness of the outcome distribution in that population after implementation. Further, it is unclear how using a model that already assumes that independent variables may affect the variability of the outcome (by parameterizing this relationship) can alone provide empirical support for the that notion. Instead, having used such a model, the authors could report on the effect estimates of the risk factor on the variability of the outcome measures. In other words, more clarity is needed regarding the takeaway message of the manuscript.

Thank you for these comments. We have edited the manuscript to clarify the takeaway message. It serves to be a tutorial for the use and interpretation of GAMLSS, and uses empirical examples which are chosen to be both novel and of substantive interest (thereby increasing the motivation for the tutorial content). Please see the revised introduction.

We suggest that the authors make this manuscript a tutorial; if they agree with our suggestion, the following additions would considerably improve the manuscript:i) Clearly annotated R and Stata codes to replicate the analysis. This would provide potential users of the proposed technique t with hands-on exercise.

As suggested this is now provided in the form of a 1) a general tutorial for using GAMLSS and associated R packages (R syntax only as Stata does not support GAMLSS); 2) annotated syntax to replicate in full the analyses conducted in this manuscript.

ii) Clear examples of interpretation within epidemiological context. For example, how should one interpret the percentage point difference in SD and the uncertainty around it?

We have provided more details on the measures of variability in order to aid lay understanding (Methods, Analytical strategy paragraphs 3-4). The new Figure 1 provides a visual depiction of distributions which differ in variability to aid this.

iii) Comparison between the results of GAMLSS and a technique that does not model the variance and further elaboration on the advantages of fitting this complex model over a simple model.

We have included this. Linear regression results would only investigate mean differences; please see the introduction (paragraph 2), Methods, Analytical strategy paragraphs 1-2; and results of mean differences shown in Tables 1 and 2 which would match those from linear regression results. Results show that GAMLSS enables important inferences to be drawn to which more simple modelling of means (linear regression) or binary outcomes (logistic regression) do not.

iv) Explanations answering the following questions: What do we learn from comparing the descriptive kernel density estimates to the unadjusted estimates? Are they supposed to be very similar? If yes, why?

We have clarified that these were created with the intention of being identical, to aid interpretation of the more complex GAMLSS analyses (Methods, Analytical strategy paragraph 1).

v) Discussions on or recommendation for addressing the on challenges in choosing the type of outcome distribution in GAMLSS within epidemiological context.

We now included our recommendation of two distributions for use in epidemiological research (Methods, Analytical strategy paragraph 2-3).